# LEARNING HUMAN POSTURAL CONTROL WITH HIER-ARCHICAL ACQUISITION FUNCTIONS

## ABSTRACT

Learning control policies in robotic tasks requires a large number of interactions due to small learning rates, bounds on the updates or unknown constraints. In contrast humans can infer protective and safe solutions after a single failure or unexpected observation. In order to reach similar performance, we developed a hierarchical Bayesian optimization algorithm that replicates the cognitive inference and memorization process for avoiding failures in motor control tasks. A Gaussian Process implements the modeling and the sampling of the acquisition function. This enables rapid learning with large learning rates while a mental replay phase ensures that policy regions that led to failures are inhibited during the sampling process. The features of the hierarchical Bayesian optimization method are evaluated in a simulated and physiological humanoid postural balancing task. We quantitatively compare the human learning performance to our learning approach by evaluating the deviations of the center of mass during training. Our results show that we can reproduce the efficient learning of human subjects in postural control tasks which provides a testable model for future physiological motor control tasks. In these postural control tasks, our method outperforms standard Bayesian Optimization in the number of interactions to solve the task, in the computational demands and in the frequency of observed failures.

## 1 INTRODUCTION

Autonomous systems such as anthropomorphic robots or self-driving cars must not harm cooperating humans in co-worker scenarios, pedestrians on the road or them selves. To ensure safe interactions with the environment state-of-the-art robot learning approaches are first applied to simulations and afterwards an expert selects final candidate policies to be run on the real system. However, for most autonomous systems a fine-tuning phase on the real system is unavoidable to compensate for unmodelled dynamics, motor noise or uncertainties in the hardware fabrication.

Several strategies were proposed to ensure safe policy exploration. In special tasks like in robot arm manipulation the operational space can be constrained, for example, in classical null-space control approaches Baerlocher & Boulic (1998); Slotine (1991); Choi & Kim (2000); Gienger et al. (2005); Saab et al. (2013); Modugno et al. (2016) or constraint black-box optimizer Hansen et al. (2003); Wierstra et al. (2008); Kramer et al. (2009); Sehnke et al. (2010); Arnold & Hansen (2012). While this null-space strategy works in controlled environments like research labs where the environmental conditions do not change, it fails in everyday life tasks as in humanoid balancing where the priorities or constraints that lead to hardware damages when falling are unknown.

Alternatively, limiting the policy updates by applying probabilistic bounds in the robot configuration or motor command space Bagnell & Schneider (2003); Peters et al. (2010); Rueckert et al. (2014); Abdolmaleki et al. (2015); Rueckert et al. (2013) were proposed. These techniques do not assume knowledge about constraints. Closely related are also Bayesian optimization techniques with modulated acquisition functions Gramacy & Lee (2010); Berkenkamp et al. (2016); Englert & Toussaint (2016); Shahriari et al. (2016) to avoid exploring policies that might lead to failures. However, all these approaches do not avoid failures but rather an expert interrupts the learning process when it anticipates a potential dangerous situation.

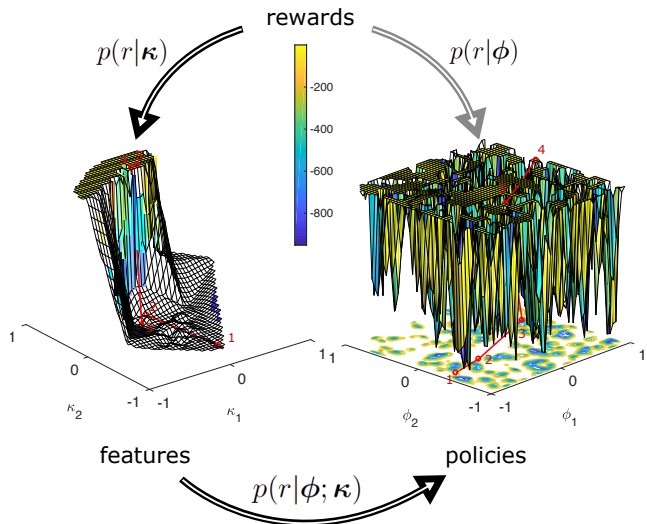

Figure 1: Illustration of the hierarchical BO algorithm. In standard BO (clock-wise arrow), a mapping from policy parameters to rewards is learned, i.e., $\phi \mapsto r \in \mathbb{R}^1$. We propose a hierarchical process, where first features $\kappa$ are sampled and later used to predict the potential of policies conditioned on these features, $\phi|\kappa \mapsto r$. The red dots show the first five successive roll-outs in the feature and the policy space of a humanoid postural control task.

All the aforementioned strategies cannot avoid harming the system itself or the environment without thorough experts knowledge, controlled environmental conditions or human interventions. As humans require just few trials to perform reasonably well, it is desired to enable robots to reach similar performance even for high-dimensional problems. Thereby, most approaches are based on the assumption of a "low effective dimensionality", thus most dimensions of a high-dimensional problem do not change the objective function significantly. In Chen et al. (2012) a method for relevant variable selection based on Hierarchical Diagonal Sampling for both, variable selection and function optimization, has been proposed. Randomization combined with Bayesian Optimization is proposed in Wang et al. (2013) to exploit effectively the aforementioned "low effective dimensionality". In Li et al. (2018) a dropout algorithm has been introduced to overcome the high-dimensionality problem by only train onto a subset of variables in each iteration, evaluating a "regret gap" and providing strategies to reduce this gap efficiently. In Rana et al. (2017) an algorithm has been proposed which optimizes an acquisition function by building new Gaussian Processes with sufficiently large kernel-lengths scales. This ensures significant gradient updates in the acquisition function to be able to use gradient-dependent methods for optimization.

The contribution of this paper is a computational model for psychological motor control experiments based on hierarchical acquisition functions in Bayesian Optimization (HiBO). Our motor skill learning method uses features for optimization to significantly reduce the number of required roll-outs. In the feature space, we search for the optimum of the acquisition function by sampling and later use the best feature configuration to optimize the policy parameters which are conditioned on the given features, see also Figure 1. In postural control experiments, we show that our approach reduces the number of required roll-outs significantly compared to standard Bayesian Optimization. The focus of this study is to develop a testable model for psychological motor control experiments where well known postural control features could be used. These features are listed in Table 3. In future work we will extend our model to autonomous feature learning and will validate the approach in more challenging robotic tasks where 'good' features are hard to hand-craft.

## 2 METHODS

In this section we introduce the methodology of our hierarchical BO approach. We start with the general problem statement and afterwards briefly summarize the concept of BO which we use here

as a baseline. We then go through the basic principles required for our algorithm and finally we explain mental replay.

## 2.1 PROBLEM STATEMENT

The goal in contextual policy search is to find the best policy $\pi^*(\boldsymbol{\theta}|\boldsymbol{c})$ that maximizes the return

$$J(\boldsymbol{\theta}) = \mathbb{E}\left[\sum_{t=0}^{T} \{r_t(\boldsymbol{x}_t, \boldsymbol{u}_t) \,|\, \pi(\boldsymbol{\theta}|\boldsymbol{c})\}\right]\,, \tag{1}$$

with reward $r_t(\boldsymbol{x}_t, \boldsymbol{u}_t)$ at time step $t$ for executing the motor commands $\boldsymbol{u}_t$ in state $\boldsymbol{x}_t$.

For learning the policy vector and the context, we collect samples of the return $J(\boldsymbol{\theta}^{[k]}) \in \mathbb{R}^1$, the evaluated policy parameter vector $\boldsymbol{\theta}^{[k]} \in \mathbb{R}^m$ and the observed contextual features modeled by $\boldsymbol{c}^{[k]} \in \mathbb{R}^n$. All variables used are summarized in Table 1. The optimization problem is defined as

$$\langle \boldsymbol{\theta}^*, \boldsymbol{c}^* \rangle = \operatorname*{argmax}_{\boldsymbol{\theta}, \boldsymbol{c}} \mathbb{E}\left[\,J(\boldsymbol{\theta}) \,|\, \pi(\boldsymbol{\theta}|\boldsymbol{c})\,\right]\,. \tag{2}$$

The optimal parameter vector and the corresponding context vector can be found in an hierarchical optimization process which is discussed in Section 2.3.

## 2.2 BAYESIAN OPTIMIZATION (BASELINE)

Bayesian Optimization (BO) has emerged as a powerful tool to solve various global optimization problems where roll-outs are expensive and a sufficient accurate solution has to be found with only few evaluations, e.g. Lizotte et al. (2007); Martinez-Cantin et al. (2007); Calandra et al. (2016). In this paper we use the standard BO as a benchmark for our proposed hierarchical process. Therefore, we now briefly summarize the main concept. For further details refer to Shahriari et al. (2016).

The main concept of Bayesian Optimization is to build a model for a given system based on the so far observed data $D = \{X, \boldsymbol{y}\}$. The model describes a transformation from a given data point $\boldsymbol{x} \in X$ to a scalar value $y \in \boldsymbol{y}$, e.g. from the parameter vector $\boldsymbol{\theta}$ to the return $J(\boldsymbol{\theta})$. Such model can either be parametrized or non-parametrized and is used for choosing the next query point by evaluating an acquisition function $\alpha(D)$. Here, we use the non-parametric Gaussian Processes (GPs) for modeling the unknown system which are state-of-the-art model learning or regression approaches Williams & Rasmussen (1996; 2006) that were successfully used for learning inverse dynamics models in robotic applications Nguyen-Tuong et al. (2009); Calandra et al. (2015). For comprehensive discussions we refer to Rasmussen (2003); Nguyen-Tuong & Peters (2011).

GPs represent a distribution over a partial observed system in the form of

$$\begin{bmatrix} \boldsymbol{y} \\ \boldsymbol{y}_* \end{bmatrix} \sim N\left(\begin{bmatrix} \boldsymbol{m}(X) \\ \boldsymbol{m}(X_*) \end{bmatrix}, \begin{bmatrix} \boldsymbol{K}(X, X) & \boldsymbol{K}(X, X_*) \\ \boldsymbol{K}(X_*, X) & \boldsymbol{K}(X_*, X_*) \end{bmatrix}\right), \tag{3}$$

where $D = \{X, \boldsymbol{y}\}$ are the so far observed data points and $D_* = \{X_*, \boldsymbol{y}_*\}$ the query points. This representation is fully defined by the mean $\boldsymbol{m}$ and the covariance $\boldsymbol{K}$. We chose $m(\boldsymbol{x}) = 0$ as mean

Table 1: Variable definitions used in this paper.

| | | |
|---|---|---|
| $J(\boldsymbol{\theta})$ | $\mathbb{R}^1$ | return of a rollout |
| $r_t(\boldsymbol{x}_t, \boldsymbol{u}_t)$ | $\mathbb{R}^1$ | intermediate reward give at time $t$ |
| $\boldsymbol{x}_t$ | $\mathbb{R}^d$ | state of the system |
| $\boldsymbol{u}_t$ | $\mathbb{R}^l$ | motor commands of the system |
| $\pi(\boldsymbol{\theta}|\boldsymbol{c})$ | | unknown control policy |
| $\boldsymbol{\theta}$ | $\mathbb{R}^m$ | policy vector |
| $\boldsymbol{c}$ | $\mathbb{R}^n$ | context vector |
| $s^{[k]}$ | $\{0, 1\}$ | flag indicating the success of rollout $k$ |

function and as covariance function a *Matérn kernel* Matérn (1960). It is a generalization of the *squared-exponential kernel* that has an additional parameter $\nu$ which controls the smoothness of the resulting function. The smoothing parameter can be beneficial for learning local models. We used *Matérn kernels*

$$k(\boldsymbol{x}_p, \boldsymbol{x}_q) = \sigma^2 \frac{1}{2^{\nu-1}\Gamma(\nu)} A^\nu \, \mathrm{H}_\nu \, A + \sigma_y^2 \, \delta_{pq},$$

with $\nu = 5/2$ and the gamma function $\Gamma$, $A = (2\sqrt{\nu}||\boldsymbol{x}_p - \boldsymbol{x}_q||)/l$ and modified *Bessel* function $\mathrm{H}_\nu$ Abramowitz & Stegun (1965). The length-scale parameter of the kernel is denoted by $\sigma$, the variance of the latent function is denoted by $\sigma_y$ and $\delta_{pq}$ is the *Kronecker* delta function (which is one if $p = q$ and zero otherwise). Note that for $\nu = 1/2$ the *Matérn kernel* implements the *squared-exponential kernel*. In our experiments, we optimized the hyperparameters $\boldsymbol{\theta} = [\sigma, l, \sigma_y]$ by maximizing the marginal likelihood Williams & Rasmussen (2006).

Predictions for a query points $D_* = \{\boldsymbol{x}_*, y_*\}$ can then be determined as

$$\begin{aligned}
\mathrm{E}(y_*|\boldsymbol{y}, X, \boldsymbol{x}_*) &= \mu(\boldsymbol{x}_*) = m_* + \boldsymbol{K}_*^\top \boldsymbol{K}^{-1}(\boldsymbol{y} - \boldsymbol{m}) \\
\mathrm{var}(y_*|\boldsymbol{y}, X, \boldsymbol{x}_*) &= \sigma(\boldsymbol{x}_*) = \boldsymbol{K}_{**} - \boldsymbol{K}_*^\top \boldsymbol{K}^{-1}\boldsymbol{K}_*.
\end{aligned} \tag{4}$$

The predictions are then used for choosing the next model evaluation point $\boldsymbol{x}_n$ based on the acquisition function $\alpha(\boldsymbol{x}; D)$. We use expected improvement (EI) Mockus et al. (1978) which considers the amount of improvement

$$\begin{aligned}
\alpha(\boldsymbol{x}; D) = {}&(\mu(\boldsymbol{x}) - \tau)\,\Phi\left(\frac{\mu(\boldsymbol{x}) - \tau + \xi}{\sigma(\boldsymbol{x})}\right) \\
&+ \sigma(\boldsymbol{x})\phi\left(\frac{\mu(\boldsymbol{x}) - \tau + \xi}{\sigma(\boldsymbol{x})}\right),
\end{aligned} \tag{5}$$

where $\tau$ is the so far best measured value $\max(\boldsymbol{y})$, $\Phi$ the standard normal cumulative distribution function, $\phi$ the standard normal probability density function and $\xi \sim \sigma_\xi U(-0.5, 0.5)$ a random value to ensure a more robust exploration. Samples, distributed over the area of interest, are evaluated and the best point is chosen for evaluation based on the acquisition function values.

## 2.3 HIERARCHICAL SAMPLING FROM ACQUISITION FUNCTIONS IN BAYESIAN OPTIMIZATION

We learn a joint distribution $p(J(\boldsymbol{\theta}^{[k]}), \boldsymbol{\theta}^{[k]}, \boldsymbol{c}^{[k]})$ over $k = 1, ..., K$ roll-outs of observed triples. This distribution is used for as a prior of the acquisition function in Bayesian optimization. However, instead of directly conditioning on the most promising policy vectors using $\alpha_{BO} = \alpha(\boldsymbol{\theta}; D)$, we propose an iterative conditioning scheme. Therefore, the two acquisition functions

$$\alpha_{\boldsymbol{c}} = \alpha(\boldsymbol{c}; D), \tag{6}$$

$$\alpha_{\boldsymbol{\theta}} = \alpha(\boldsymbol{\theta}; \boldsymbol{c}, D), \tag{7}$$

are employed, where for Equation (7), the evaluated mean $\mu(\boldsymbol{\theta}; \boldsymbol{c})$ and variance $\sigma(\boldsymbol{\theta}; \boldsymbol{c})$ for the parameter $\boldsymbol{\theta}$ are conditioned on the features $\boldsymbol{c}$. The hierarchical optimization process works then as follows:

In the first step we estimate the best feature values based on a GP model using the acquisition function from Equation (6)

$$\boldsymbol{c}^{[k+1]} = \max_{\boldsymbol{c}} \alpha(\boldsymbol{c}; D^{[1:k]}). \tag{8}$$

These feature values are then used to condition the search for the best new parameter $\boldsymbol{\theta}^{[k+1]}$ using Equation (7)

$$\boldsymbol{\theta}^{[k+1]} = \max_{\boldsymbol{\theta}} \alpha(\boldsymbol{\theta}; \boldsymbol{c}^{[k+1]}, D^{[1:k]}). \tag{9}$$

We subsequently continue evaluating the policy vector $\boldsymbol{\theta}^{[k+1]}$ using the reward function presented in Equation (1). Finally, the new data point $\langle J(\boldsymbol{\theta}^{[k+1]}), \boldsymbol{\theta}^{[k+1]}, \boldsymbol{c}^{[k+1]} \rangle$ can be added to the set of data points $D$.

---

**Algorithm 1** Hierarchical Acquisition Function Sampling for Bayesian Optimization (HiBO)

---

1: Initialize the dataset $D^{[1:k]} = \langle J(\boldsymbol{\theta}^{[k]}), \boldsymbol{\theta}^{[k]}, \boldsymbol{c}^{[k]} \rangle$ with $K$ rollouts of sampled policies $\boldsymbol{\theta}$.
2: **for** k = K, K+1, ... **do**
3:     $\boldsymbol{c}^{[k+1]} = \text{argmax}_{\boldsymbol{c}} \, \alpha(D^{[1:k]}) : D \rightarrow \mathbb{R}^1$ using Eq. 6.
4:     $\boldsymbol{\theta}^{[k+1]} = \text{argmax}_{\boldsymbol{\theta}} \, \alpha(D^{[1:k]}; \boldsymbol{c}^{[k+1]})$ using Eq. 7.
5:     Evaluate the policy vector $\boldsymbol{\theta}^{[k+1]}$ using Eq. 1.
6:     Augment $D = [D^{[1:k]}, \langle J(\boldsymbol{\theta}^{[k+1]}), \boldsymbol{\theta}^{[k+1]}, \boldsymbol{c}^{[k+1]} \rangle^l]$.
7: **end for**

---

## 2.4 MENTAL REPLAY

To ensure robustness for Bayesian Optimization, mental replays can be generated. Therefore, the new training data set $\langle J(\boldsymbol{\theta}^{[k+1]}), \boldsymbol{\theta}^{[k+1]}, \boldsymbol{c}^{[k+1]} \rangle$, generated by the policy parameter $\boldsymbol{\theta}^{[k+1]}$, will be enlarged by augmenting perturbed copies of the policy parameter $\boldsymbol{\theta}^{[k+1]}$. These $l$ copies are then used for generating the augmented training data sets

$$D^{[k+1]} = \langle J(\boldsymbol{\theta}^{[k+1]}), \boldsymbol{\theta}^{[k+1]}, \boldsymbol{c}^{[k+1]} \rangle^l. \tag{10}$$

Here, the transcript $\langle \cdot \rangle^l$ denotes $l$ perturbed copies of the given set. Hence, perturbed copies of the parameters $\boldsymbol{\theta}^{[k+1]}$ and features $\boldsymbol{c}^{[k+1]}$ are generated keeping the objective $J(\boldsymbol{\theta}^{[k+1]})$ constant. In Algorithm (1) the complete method is summarized. We evaluate different replay strategies in the result Section in 3.3.

## 3 RESULTS

In this section we first present observations on human learning during perturbed squat-to-stand movements. We compare the learning results of a simulated humanoid to the learning rates achieved by the human participants. Second, we evaluate our hierarchical BO approach in comparison to our baseline, the standard BO. Third we evaluate the impact of mental replays on the performance of our algorithm.

### 3.1 HUMAN POSTURAL BALANCING

To observe human learning, we designed an experiment where 20 male participants were subjected to waist pull perturbation during squat-to-stand movements, see Figure 2(a). Participants had to stand up from a squat position without making any compensatory steps (if they made a step, such trial was considered a fail). Backward perturbation to the centre of mass (CoM) was applied by a pulling mechanism and was dependent on participants' mass and vertical CoM velocity.

On average, participants required 6 trials ($\sigma_{\text{human}} = 3.1$) to successfully complete the motion. On the left panel of Figure 3, a histogram of the required trials before the first success is shown. On the right panel, the evaluation results for the simulated humanoid are presented (details on the implementation are discussed in the subsequent Subsection 3.2). The human learning behavior is faster and more reliable than the learning behavior of the humanoid. However, humans can exploit fundamental knowledge about whole body balancing whereas our humanoid has to learn everything from scratch. Only the gravity constant was set to zero in our simulation, as we are only interested in the motor adaptation and not in gravity compensation strategies.

Adaptation was evaluated using a measure based on the trajectory area (TA) at every episode as defined in **?**. The Trajectory area represents the total deviation of the CoM trajectory with respect to a straight line. The trajectory area of a given perturbed trajectory is defined as the time integral of the distance of the trajectory points to the straight line in the sagittal plane:

$$TA(e_x) = \int_{t_0}^{t_{end}} x(t)|\dot{y}(t)|\mathrm{d}t \tag{11}$$

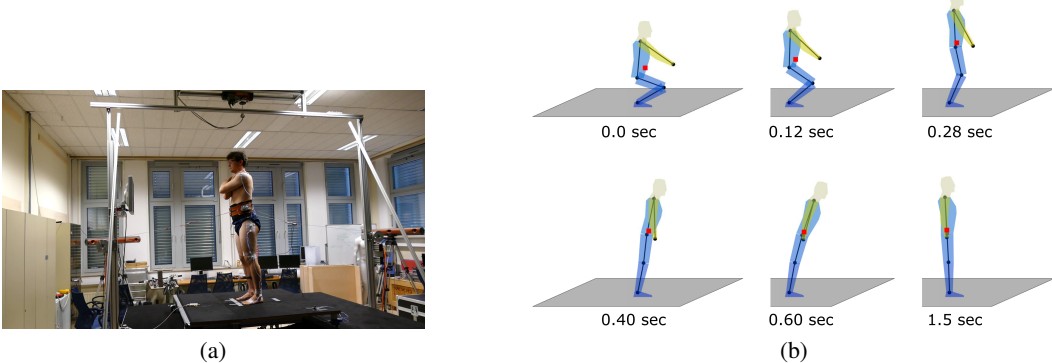

(a)                            (b)

Figure 2: (a) Psychological postural control setup for the squat-to-stand movements. (b) Illustration of the simulated postural control task. An external perturbation is applied during the standing up motion and the robot has to learn to counter balance. The perturbation is proportional to the CoM velocity in the superior direction in the *sagittal plane*.

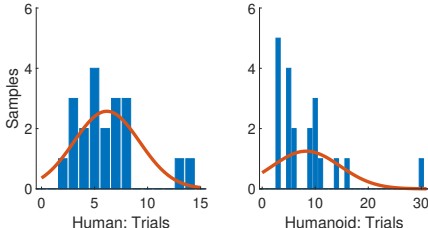

Figure 3: Histogram showing the number of required trials until the first successful episode for both, the human experiments and the simulated humanoid, with $\mu_{\text{human}} = 6.15$, $\sigma_{\text{human}} = 3.1$, $\mu_{\text{humanoid}} = 8.3$ and $\sigma_{\text{humanoid}} = 6.38$.

A positive sign represents the anterior direction while a negative sign represents the posterior direction. The mean and standard deviation for the trajectory area over the number of training episodes for all participants are depicted in Figure 4. Comparing these results with the simulation results of our humanoid shows that the learning rate using our approach is similar to the learning rate of real humans.

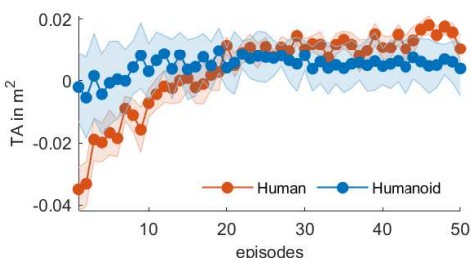

Figure 4: Mean and standard deviaton of the trajectory area ($TA$) with regard to the number of episodes for both, the human experiments and the simulated humanoid. For the humanoid the $x$-coordinates have been shifted about $-0.5$ to account for the stretched arms. In addition, the trajectory area of the humanoid has been scaled with the factor $0.1$ and shifted about $-0.2$ to allow easier comparison.

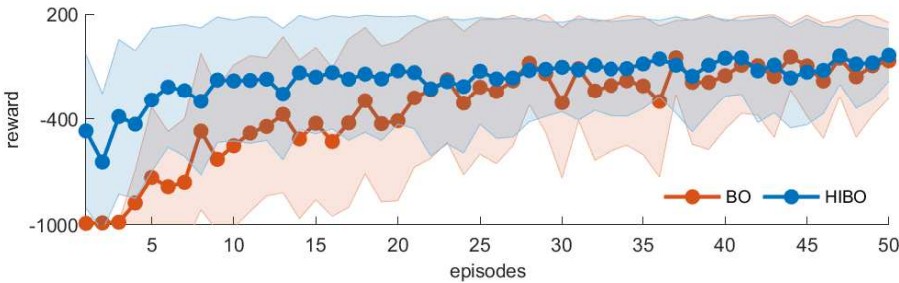

Figure 5: Comparison of the rewards of the proposed *HIBO* algorithm and the state-of-the-art approach *Bayesian Optimization*. Shown are average statistics (mean and standard deviation) over $20$ runs.

## 3.2 HUMANOID POSTURAL BALANCING

To test the proposed algorithm we simulated a humanoid postural control task as shown in Figure 2(b). The simulated humanoid has to stand up and is thereby exposed to an external pertubation proportional to the velocity of the CoM in the superior direction in the sagittal plane. The pertubation is applied during standing up motion such that the robot has to learn to counter balance. The simulated humanoid consist of four joints, connected by rigid links, where the position of the first joint is fixed onto the ground. A PD-controller is used with $K_{P,i}$ and $K_{D,i}$ for $i = 1, 2, 3, 4$ being the proportional and derivative gains. In our simulations the gains are set to $K_{P,i} = 400$ and $K_{D,i} = 20$ and an additive control noise $\epsilon \sim \mathcal{N}(0, 1)$ has been inserted such that the control input for a certain joint becomes

$$u_i = K_{P,i} \, e_{P,i} + K_{D,i} \, e_{D,i} + \epsilon, \tag{12}$$

where $e_{P,i}$, $e_{D,i}$ are the joint errors regarding the target position and velocity. The control gains can also be learned. The goal positions and velocities for the joints are given. As parametrized policy, we use a via point $[\phi_i, \dot{\phi}_i]$, where $\phi_i$ is the position of joint $i$ at time $t_{\text{via}}$ and $\dot{\phi}_i$ the corresponding velocity. Hence, the policy is based on 9, respectively 17 parameters (if the gains are learned), which are summarized in Table 2. For our simulations we handcrafted 7 features, namely the overall success, the maximum deviation of the CoM in $x$ and $y$ direction and the velocities of the CoM for the $x$ and $y$ directions at $200\,ms$ respectively $400\,ms$. In Table 3 the features used in this paper are summarized. Simultaneously learning of the features is out of scope of this comparison to human motor performance but part of future work.

We simulated the humanoid in each run for a maximum of $t_{\max} = 2\,s$ with a simulation time step of $dt = 0.002\,s$, such that a maximum of $N = 1000$ simulation steps are used. The simulation has been stopped at the simulation step $N_{\text{end}}$ if either the stand up has been failed or the maximum simulation time has been reached. The return of a roll-out $J(\boldsymbol{\theta})$ is composed according to $J(\boldsymbol{\theta}) = -(c_{\text{balance}} + c_{\text{time}} + c_{\text{control}})$ with the balancing costs $c_{\text{balance}} = 1/N_{\text{end}} \sum_{i=1}^{N_{\text{end}}} ||\boldsymbol{x}_{\text{CoM,target}} - \boldsymbol{x}_{\text{CoM,i}}||^2$, the time dependent costs $c_{\text{time}} = (N - N_{\text{end}})$ and control costs of $c_{\text{control}} = 10^{-8} \sum_{i=1}^{N_{\text{end}}} \sum_{j=1}^{4} u_{ij}^2$.

We compared our approach with our baseline, standard Bayesian Optimization. For that we used the features $4, 5$ in $3$ and set the number of mental replays to $l = 3$. We initialized both, the BO and the HiBO approach with 3 seed points and generated average statistics over 20 runs. In Figure 5 the comparison between the rewards of the algorithms over 50 episodes is shown. In Figure 6 (a) the

Table 2: Policy parameter description

| | |
|---|---|
| $K_{P,i}$ | proportional gain for joint $i$ |
| $K_{D,i}$ | derivative gain for joint $i$ |
| $\phi_i$ | angle of joint $i$ at the via point |
| $\dot{\phi}_i$ | angular velocity of joint $i$ at the via point |
| $t_{\text{via}}$ | time for switching from the via point to goal position |

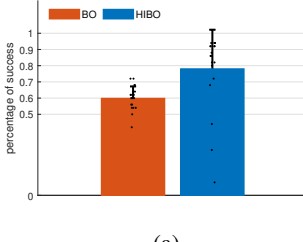 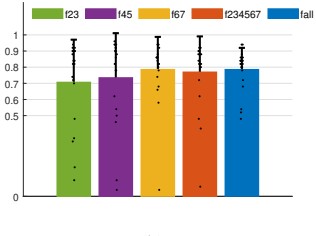 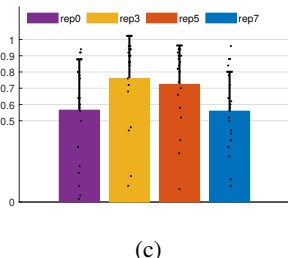

|(a)|(b)|(c)|

Figure 6: Comparison of the number of successful episodes of the proposed *HIBO* algorithm and the state-of-the-art approach *Bayesian Optimization* for different internal experience replay iterations. The last three algorithms implemented an *automatic relevance determination* of the *Gaussian Process* features and policy parameters. Shown are average statistics (mean and standard deviation) over 20 runs and the true data values are denoted by the black dots.

number of successful episodes is illustrated. Our approach requires significantly fewer episodes to improve the reward then standard Bayesian Optimization ($10 \pm 3$ vs $45 \pm 5$) and has a higher success quote ($78\% \pm 24\%$ vs $60\% \pm 7\%$).

We further evaluated the impact of the different features on the learning behavior. In Figure 6 (b) the average statistics over 20 runs for different selected features with 3 mental replays are shown. All feature pairs generate better results on average then standard BO, whereas for the evaluated task no significant difference in the feature choice was observed.

## 3.3 EXPLOITING MENTAL REPLAYS

We evaluated our approach with additional experience replays. For that we included an additive noise of $\epsilon_{\text{rep}} \sim \mathcal{N}(0, 0.05)$ to perturb the policy parameters and features. In Figure 6 (c) average statistics over 20 runs of the success rates for different number of replay episodes are shown (rep3 = 3 replay episodes). Our proposed algorithm works best with a number of 3 replay episodes. Five or more replays in every iteration steps even reduce the success rate of the algorithm.

## 4 CONCLUSION

We introduced HiBO, a hierarchical approach for Bayesian Optimization. We showed that HiBO outperforms standard BO in a complex humanoid postural control task. Moreover, we demonstrated the effects of the choice of the features and for different number of mental replay episodes. We compared our results to the learning performance of real humans at the same task. We found that the learning behavior is similar. We found that our proposed hierarchical BO algorithm can reproduce the rapid motor adaptation of human subjects. In contrast standard BO, our comparison method, is about four times slower. In future work, we will examine the problem of simultaneously learning task relevant features in neural nets.

Table 3: Feature description

| | |
|---|---|
| Feature 1 | success |
| Feature 2 | maximum deviation of the CoM in $x$ direction |
| Feature 3 | maximum deviation of the CoM in $y$ direction |
| Feature 4 | velocity of the CoM in $x$ direction at $200\,ms$ |
| Feature 5 | velocity of the CoM in $y$ direction at $200\,ms$ |
| Feature 6 | velocity of the CoM in $x$ direction at $400\,ms$ |
| Feature 7 | velocity of the CoM in $y$ direction at $400\,ms$ |

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
