# OpenReview forum: "Learning Human Postural Control with Hierarchical Acquisition Functions"
_ICLR.cc/2020/Conference — Reject_

### Official Review · AnonReviewer1 · 2019-10-21
**Official Blind Review #1**

**Rating:** 1

**Review:**

How to quickly learn control policies with minimized number of environment interactions have long been an important problem. To tackle this problem, this paper proposed a "hierarchical Bayesian optimization (HIBO)" algorithm to optimize the "feature parameter \phi" (which I don't know what that is) and the "policy parameter \theta" hierarchically. Under the formulation of maximizing reward J(\theta), the algorithm firstly uses EI to select \phi. Given the selected \phi, the algorithm selects the policy parameter \theta. The proposed algorithm is evaluated on a Humanoid Postural Balancing task, and shows achieves high rewards faster than the standard EI acquisition. However, the paper is awfully written such that I cannot understand what the "feature parameter \phi" is. Given my limited understanding, I think the paper should be rejected.

Strengths,
1, The paper deals with an interesting task: Humanoid Postural Balancing. A Humanoid is expected to learn how to balance as quick as possible to reduce the interactions with the environments, which suits well with Bayesian optimization.

Weakness,
1, The paper is awfully written. The problem statement subsection is unreadable. I don't see anywhere explaining how the states x_t, commands u_t, \theta, and feature \phi are related? What is \phi ? It is super wired why the feature parameter \phi is jointly maximized with the policy parameter.  Because I don't understand the formulation, I can hardly understand anything else.
2, From my very limited understanding on the formulation, the proposed HIBO is trivial.
3, The experiments are limited. The paper only conducts one experiment on the Humanoid control balancing. And they paper only compares with the EI acquisition, while the state-of-art acquisition MES should be also be compared with.
4, The proposed mental replay is not well justified, qualitatively or empirically.

**Experience Assessment:**

I have published one or two papers in this area.

**Review Assessment: Checking Correctness Of Derivations And Theory:**

N/A

**Review Assessment: Checking Correctness Of Experiments:**

I assessed the sensibility of the experiments.

**Review Assessment: Thoroughness In Paper Reading:**

N/A

---

> ### Author Response · Authors · 2019-11-11
> **Reply to the Official Blind Review #1**
>
> Dear reviewer,
>
> thank you for taking a look at our paper. However,
>
> 1) all variables used in the paper are clearly defined and listed (e.g., in Table 1). In page 7, after Equation 12, the concrete policy implementation for the task is discussed.
>
> 2) "1, The paper is awfully written.”
> It is not sufficient to publish such a statement without concrete references. If you refer to point 3) then we strongly disagree.
>
> 3) "I don't see anywhere explaining how the states x_t, commands u_t, \theta, and feature \phi are related? “
> In Equation 1, the mentioned variables are put in relation. Moreover, problem statements of our form are common in reinforcement learning and machine learning. Given your expertise (Experience Assessment: I have published one or two papers in this area.) this should be known.
>
> 4) "What is \phi ? It is super wired why the feature parameter \phi is jointly maximized with the policy parameter. Because I don't understand the formulation, I can hardly understand anything else.”
> Please see our answer to point 1). We discuss a standard optimization problem definition which is well known in the reinforcement learning and machine learning community.
>
> 5) "From my very limited understanding on the formulation, the proposed HIBO is trivial.”
> Can you provide further details on that statement or is this speculative?
>
> 6) "The experiments are limited. The paper only conducts one experiment on the Humanoid control balancing. And they paper only compares with the EI acquisition, while the state-of-art acquisition MES should be also be compared with.”
> We evaluated our approach on a challenging non-linear postural control task. We compared to the most closely related approach that is Bayesian optimization and evaluated several acquisition functions and kernels (EI, UCB, EI+ARD, UCB+ARD). We did not observe any statistical significant difference and only reported the results for EI. However, we will add a summary statement to the results and details to a supplement.
> It is important to note that our proposed hierarchical acquisition function can be implemented with any acquisition function (EI, LCB, PI, MES, etc.). However, we thank the reviewer for the link to [1] and will also eval this acquisition function.
>
> 7) "The proposed mental replay is not well justified, qualitatively or empirically.”
> The implemented mental replay is a common and well known practice in reinforcement learning. We will add references to related work.
>
> Our questions to the reviewer:
>
> A1) Your "Review Assessment: Checking Correctness Of Derivations And Theory: N/A” was selected because of your limited experience in the field?
>
> A2) How can we interpret your input on "Review Assessment: Thoroughness In Paper Reading: N/A”? You had limited time to review the paper?
>
>
> [1] Wang, Zi, and Stefanie Jegelka. "Max-value entropy search for efficient Bayesian optimization." Proceedings of the 34th International Conference on Machine Learning-Volume 70. JMLR. org, 2017.

---

> > ### Comment · AnonReviewer1 · 2019-11-13
> > **Reply**
> >
> > 1,2,3,4) Firstly, I apologize for using the word "awfully", which might give you a hard time. I respect your hard work, but reading the paper also gave me a hard time.
> >
> > The problem is that the meanings of your jargons are not explained.
> >
> > For example, the "context" vector $c$, if you look at Eq(1), how $c$ is used besides being conditioned by $\theta$. But $\theta$ is the parameter to be optimized, so what's the functionality of $c$? On the other hand, if you could have an example for the setup, it will make more sense. But even in the experiment section, I don't see what is the context "c".
> >
> > Furthermore, for what "is well known in the reinforcement learning and machine learning community", the notation $\pi$ is used for: "$\pi(a|s)$ is the probability that action is $a$ if state is $s$" (from Page 58, Richard Sutton's RL book). It took me a long time to figure out why do you have two actions "\theta" and $u$.
> >
> > 5) My novelty judgement is speculatively guess based on my experiences. I have stressed that it is based on "my very limited understanding". Because the proposed approach is mostly related to the context $c$ and the parameter $\theta$, it will be hard to understand before understanding what is $c$.
> >
> > 6) For the experiment section in the main article, it should be focused on evaluating the proposed method instead of talking about the experiment details. However, it is until Figure 5 that the proposed HIBO is compared to baselines. Although the experiment section has a lot of figures, useful figures are not much, that's why I gave the judgement.
> >
> > 7) I don't know the literatures about "perturbed mental replay", please tell me if I was ignorant. On the other hand, if the method is existing, it should be cited when it is introduced. Otherwise, the reader will think that it is original.
> >
> > A1) I selected that because the paper doesn't really have derivations and theories. Before section 2.3, there are existing formulas. In section 2.3, the formulas aren't new. (I am not saying the method is not new, I am just saying the "derivations" are not new).
> > A2) On the one hand, I spent a lot of time on this paper. This paper was the first one I read between all I need to review, because I thought it looked interesting. I read it twice or so but could not understand. On the time of writing reviews, I read it probably another once or twice. On the other hand, I am busy. I hope I can learn something from the reviewing process so that I don't waste my time on the reviewing. Unfortunately, I don't think I learnt much interesting stuff from reviewing this paper.
> >
> > Overall, the best thing happens when one reviewer enjoys the paper he is reading and the authors are glad with the high score from the reviewer. In the opposite, the worst thing happens.

---

### Official Review · AnonReviewer2 · 2019-10-28
**Official Blind Review #2**

**Rating:** 1

**Review:**

After rebuttal:

Thank you to the authors for responding to my review.

1) The title of the conference is "... on Learning Representations". As I stated in the review ("no, e.g., neural networks are employed"), neural networks are an *example* of, but do not subsume, all representation learning methods. Therefore, I agree that papers that do not cover neural networks are welcome at the conference. However, as I stated in the review, my evaluation of the method proposed in the submission is that it does not concern representation learning ("The employed features in Table 3 are handcrafted"). I believe this evaluation is defensible, but of course the final evaluation is up to the chairs. However, I note that the authors did not respond directly to my evaluation that the method is not engaging in representation learning.

2-7) As the other reviewer notes, the paper lacks clarity in many places, and does not sufficiently discuss prior work, including in postural control (there is one citation in the references that is not mentioned in the main text), hierarchical Bayesian optimization within or without a Gaussian processes framework (https://scholar.google.com/scholar?hl=fr&as_sdt=0%2C5&q=hierarchical+bayesian+optimization&btnG=), or experience replay (https://scholar.google.com/scholar?hl=fr&as_sdt=0%2C5&q=replay+machine+learning&btnG=). Therefore, it is difficult to ascertain the research contribution.

As such, I stand by my evaluation that this submission is not ready for publication at ICLR.

===========================

Before rebuttal:

The submission presents a hierarchical Bayesian optimization (HiBO) approach to solving a postural control task expressed as a proportional-derivative (PD) controller.

Strengths:
- The HiBO approach outperforms the non-hierarchical BO approach on the task of postural control.

Weaknesses:
- The paper does not make use of representation learning (no, e.g., neural networks are employed) and is therefore out-of-place at ICLR. The employed features in Table 3 are handcrafted.
- The task (simulating human postural control) is not well-situated in the context of prior work using HiBO for robotics, so the contribution remains unclear.
- It is not clear why this problem should be formulated as contextual policy search (i.e., to what the context variable refers).
- Only one baseline (Bayesian optimization (BO)) is reported. This baseline corresponds to the ablation of the HiBO method (i.e., the omission of the context variable), and so does not represent, more broadly, an alternative approach.
- The concept of "mental replay" is briefly introduced, but no reference is made to prior work in imagined rollouts, and no ablation study on the impact of this component is performed.

Minor points:
- It is unclear why the problem setting should be labeled as "psychological" postural control.
- There are several missing references ("?") in the text.

**Experience Assessment:**

I have read many papers in this area.

**Review Assessment: Checking Correctness Of Derivations And Theory:**

I assessed the sensibility of the derivations and theory.

**Review Assessment: Checking Correctness Of Experiments:**

I assessed the sensibility of the experiments.

**Review Assessment: Thoroughness In Paper Reading:**

I read the paper thoroughly.

---

> ### Author Response · Authors · 2019-11-11
> **Reply to the Official Blind Review #2**
>
> Dear reviewer,
>
> thank you for your review. However, we are a bit shocked about the quality of the review. In the following, we discuss your points of critics:
>
> 1) "- The paper does not make use of representation learning (no, e.g., neural networks are employed) and is therefore out-of-place at ICLR. The employed features in Table 3 are handcrafted."
> ICLR is not "just about" neural networks, as you can read here: https://iclr.cc/Conferences/2020/CallForPapers
>
> "We consider a broad range of subject areas including feature learning, metric learning, compositional modeling, structured prediction, reinforcement learning, ..."
>
> We informed the program chairs about that issue. To discredit a paper due to an obviously wrong interpretation of  the conference topics is not acceptable.
>
> 2) "The task (simulating human postural control) is not well-situated in the context of prior work using HiBO for robotics, so the contribution remains unclear."
>
> HIBO was never published prior to this submission and no prior HIBO work exists. Further, postural control is considered as one of most challenging control problems in reinforcement learning and motor control.
>
> 3) " It is not clear why this problem should be formulated as contextual policy search (i.e., to what the context variable refers)."
> The mathematical framework of contextual policy search allows us to derive a general hierarchical model that can be applied to any other contextual policy search task. The context variables denote salient features of the motor skills that we intent to learn and can be adapted if needed. Therefore, they are referred as "context variables". However, we are open for suggestions of alternative mathematical representations if you provide details.
>
> 4) "Only one baseline (Bayesian optimization (BO)) is reported. This baseline corresponds to the ablation of the HiBO method (i.e., the omission of the context variable), and so does not represent, more broadly, an alternative approach."
> BO is a state of the art approach and most closely related. It is ideally suited to evaluate the benefits or drawbacks of an extension to hierarchical acquisition functions. Note that in most contextual policy search approaches the context is fixed and assumed to be known which is not the case in our approach.
>
> 5) "- The concept of "mental replay" is briefly introduced, but no reference is made to prior work in imagined rollouts, and no ablation study on the impact of this component is performed."
> Although that mental replay is a common and well known practice in reinforcement learning, we will add references to it.
>
> 6) "- It is unclear why the problem setting should be labeled as "psychological" postural control."
> The term 'psychological' refers only to the experiment involving humans and is a standard terminology in human motor control. We will clarify that.
>
> 7) "There are several missing references ("?") in the text."
> No, there are not "several"  missing references ("?") in the text that we submitted! There is a singe missing ref "?" in page 5 to [1] due to a typo. We thank the reviewer for this comment.
>
> [1] Nakano, Eri, et al. "Quantitative examinations of internal representations for arm trajectory planning: minimum commanded torque change model." Journal of Neurophysiology 81.5 (1999): 2140-2155.

---

### Decision · Program_Chairs · 2019-12-19

**Decision:**

Reject

**Comment:**

The paper proposes hierarchical Bayesian optimization (HiBO) for learning control policies from a small number of environment interaction and applies it to the postural control of a humanoid. Both reviewers raised issues with the clarity of presentation, as well as contribution and overall fit to this venue. The authors’ response helped to clarify these issues only marginally. Therefore, primarily due to lack of clarity, I recommend rejecting this paper, but encourage the authors to improve the presentation as per the reviewers’ suggestions and resubmitting.